# Genomic characterization of antimicrobial-resistance and virulence factors in Salmonella isolates obtained from pig farms in Antioquia, Colombia

María Isabel García-Álvarez[1], Juana L. Vidal[2], Pilar Donado-Godoy[3], Jared Smith[4], Nikki Shariat[4], María Fernanda Valencia[3], Luis M. Gómez-Osorio[1], Sara López-Osorio[1], Jenny J. Chaparro-Gutiérrez[1]*

1 CIBAV Research Group, School of Veterinary Medicine, Faculty of Agrarian Sciences, University of Antioquia, UdeA, Medellín, Colombia, 2 Diagnostic Unit Faculty of Agrarian Sciences, University of Antioquia, UdeA, Medellín, Colombia, 3 Global Health Research Unit for the Genomic Surveillance of Antimicrobial Resistance-Colombia, CI Tibaitatá, Corporación Colombiana de Investigación Agropecuaria (AGROSAVIA), Mosquera, Colombia, 4 Department of Population Health, Poultry Diagnostic and Research Center, University of Georgia, Athens, Georgia, United States of America

* jenny.chaparro@udea.edu.co

## Abstract

### Background

Occurrence of antimicrobial-resistant *Salmonella* strains has been reported worldwide, because of inappropriate use of antimicrobial products in either humans or animals. The presence of multidrug resistant *Salmonella* in pig production systems had been reported in Antioquia, Colombia.

### Aim

To identify antimicrobial resistance genes (ARG) in different *Salmonella* spp. strains isolated from pig productions in Antioquia, Colombia. Methods: Samples were received at the Diagnostic Unit of the Faculty of Agrarian Sciences at the University of Antioquia, from January 1, 2019, to January 2021. A total of 28 isolates of *Salmonella* spp. were included, which presented phenotypic resistance to more than one antibiotic used in pig farms. Whole genome sequencing (WGS) was performed in the Unit of Genomic of Agrosavia using an automated pipeline from the GHRU- Sanger Institute, employing the Illumina MiSeq platform.

### Results

WGS revealed 34 ARGs among these isolates. In 25 isolates (89%) more than two ARGs were found. Genes encoding resistance were found for 10 different groups of antibiotics (beta-lactam, aminoglycosides, chloramphenicol, rifampicins, lincosamides, fluoroquinolones, tetracyclines, sulfonamides and trimethoprim). The most frequently observed MDR profile in Typhimurium isolates was AMP-CEX-CEP-CEF-EFT-CEQ-FLU-ENR-TE-FFC-SXT.

**Data availability statement:** All WGS data supporting the findings of this study have been deposited in the National Center for Biotechnology Information (NCBI) Sequence Read Archive under BioProject ID PRJNA1168507.

**Funding:** This study was supported by the Sustainability Strategy of the CIBAV Research Group, School of Veterinary Medicine, Faculty of Agricultural Sciences, University of Antioquia (UdeA), Medellín, Colombia (to JJCG), and by the Global Health Research Unit for the Genomic Surveillance of Antimicrobial Resistance-Colombia (to PDG). The funders had no role in study design, data collection and analysis, decision to publish, or preparation of the manuscript. We also acknowledge the support of the Department of Population Health, Poultry Diagnostic and Research Center, University of Georgia.

**Competing interests:** The authors have declared that no competing interests exist.

## Conclusion

The presence of multi-drug resistant *Salmonella* strains in pigs destined for human consumption in Antioquia, Colombia was determined. This research emphasizes the utmost importance of epidemiological tools to understand the presence and spreading of antimicrobial resistance genes in pig farms. Additionally, it highlights the critical need for developing educational programs and public policies to help reduce the spread of antimicrobial resistance in production systems.

## Author summary

Recent research has raised alarms about antimicrobial resistance in *Salmonella* found in pigs from farms in Antioquia, Colombia. We used whole-genome sequencing to identify 34 resistance genes in 28 Salmonella isolates, revealing that 89% of these isolates carried more than two resistance genes, affecting various antibiotics. This situation poses significant risks, as multidrug-resistant *Salmonella* in pigs could directly or indirectly impact human health and lead to potential economic losses in pig production.

Our findings highlight the urgent need to implement better management practices on farms and responsible antibiotic use in pig production. Properly addressing these issues is essential to ensure food safety and protect public health. Continuous surveillance and effective control measures are crucial to combat this growing problem. By prioritizing these actions, we can help mitigate the risks associated with antimicrobial resistance and safeguard both livestock and consumers.

## Introduction

The increased incidence of *Salmonella* spp. has a significant impact on public and animal health [1]. Human salmonellosis has been mainly associated with food animal production (beef, pork, chicken, and eggs) [2]. Globally, *Salmonella* serovar Typhimurium and its monophasic variant, serovar I 4, [3], 12:i:-, are responsible for the high incidence of human illness [4] and both serovars are commonly associated with swine.

There are nearly 10 million pigs in Colombia, 89.5% of which are from industrial production farms, while the remaining 10.5% are from smaller operations [5]. It is estimated that the consumption per capita of pork in Colombia in 2021 was 12.2 kg, and over 500 tons are produced in Colombia each year [3]. Exports for this same year reached around 100 tons of pork destined for the Ivory Coast and Hong Kong [6]. For 2021, official figures report a profit of 5,536,335 swine heads with an increase of 6.6% compared to the previous year [7]. The Antioquia region of Colombia is considered the largest producer of pork in the country and is responsible for ~11% of the swine inventory [8].

The occurrence of antimicrobial-resistant *Salmonella* infections has been reported globally [9]. Human *Salmonella* infections caused by antibiotic-resistant strains have been associated with severe disease and more significant adverse outcomes, including increased rate of hospitalizations, longer length of hospital stays, and higher mortality [10]. In swine, antimicrobial agents are critical in treating serious infections such as salmonellosis, colibacillosis, and mycoplasmal pneumonia [11]. However, in many developing countries, antimicrobials are used in swine and other food animal production for non-clinical purposes, e.g., as feed proficiency

enhancers and growth promoters [12], which, added to the ease of obtaining antibiotics in the country, has resulted in an overuse of antimicrobials [13].

Recently, antimicrobial resistance (AMR) in *Salmonella* isolates from pig production in Colombia was demonstrated, presenting multidrug resistance (resistance to ≥ 3 classes of antibiotics) in 44% of isolates [14]. These results underline how crucial it is to comprehend the dynamics of antimicrobial usage in pig production and to assess the rise of *Salmonella* resistance in Antioquia. This information is critically needed to improve judicious antimicrobial usage in the livestock sector [14,15]. Additionally, this knowledge is required to inform veterinarians, managers, and employees of piggeries on the cautious use of antibiotics to maintain pig disease susceptibility, which will benefit the consumer and may lead to a reduced production costs related to ailments brought on by AMR bacteria and their treatment [16].

This study aimed to genomically characterize antimicrobial resistance and virulence factors in *Salmonella* spp. isolates from pig productions in Antioquia, Colombia. Furthermore, a comparative analysis was conducted between phenotypic resistance profiles and their corresponding genotypic attributes.

## Materials and methods

### Endorsement of the Ethics Committee

The Ethics Committee for Animal Experimentation (CEEA) of the Universidad de Antioquia granted the ethical endorsement to perform this study according to minute No. 141 of August 3, 2021.

### Sample collection and *Salmonella* isolation

A total of 28 Salmonella spp. isolates were included. These were recovered from fecal samples, rectal swabs, and intestinal tissue samples from commercial pig farms in the department of Antioquia, Colombia. The samples were received at the Diagnostic Unit of the Faculty of Agrarian Sciences at the University of Antioquia, Colombia. The samples arrived at the laboratory routinely, and the inclusion period extended from January 1, 2019, to January 2021, during which they were preserved and processed.

### Isolation and identification

In brief, 10-25 g of feces and gut tissue samples were transferred to 225 mL of buffered peptone (BPW) water, and 9 mL of BPW was added to rectal swabs. Samples were homogenized for 2 minutes and incubated at 35 °C for 18-24 hours. For selective enrichment, 100 μL were inoculated in 10 mL of Rappaport-Vassiliadis broth, and then incubated at 42 °C for 18 to 24 hours. After incubation, broth cultures were plated on Hektoen agar and xylose lysine deoxycholate agar (XLD) and incubated at 35 °C for 18-24 hours. Blue or blue-green colonies (with or without black center) on Hektoen agar and transparent to pinkish-red colonies (with or without black center) on XLD were presumptive. These presumptive positive colonies underwent biochemical tests (TSI, LIA, citrate, and SIM) and serological tests (Check and Trace Salmonella) for Salmonella confirmation [17–19].

### Identification of presumptive bacterial isolates and antimicrobial susceptibility testing

*Salmonella* spp. isolates from pigs were sent to the Global Health Research Unit- Colombia (Corporación Colombiana de Investigación Agropecuaria (AGROSAVIA), CI Tibaitatá, Mosquera, Colombia) to be sequenced (Table 1). The presumptive *Salmonella* isolates were evaluated for identification and antibiotic susceptibility testing using the Vitek2 Compact System with GN ID and AST-N272 cards (BioMerieux, Lyon, France) according to the manufacturer's

Table 1.  Resistance patterns found in 28 *Salmonella* isolates from commercial swine farms in the Antioquia, Colombia.

| Serotype(s) | No. Of Isolates | AMP[a] | CEX[b] | CEP[c] | CEF[d] | CFPZ[e] | GEN[f] | CEQ[g] | FLU[h] | ENR[i] | TE[j] | FFC[k] | SXT[l] | MAR[m] | NEO[n] |
|---|---|---|---|---|---|---|---|---|---|---|---|---|---|---|---|
| Typhimurium | 5 | ■ | ■ | ■ | ■ | ■ |  | ■ | ■ | ■ | ■ | ■ | ■ |  |  |
| I, 4, [3],12: i: - | 3 | ■ |  |  | ■ |  |  |  | ■ |  | ■ | ■ |  |  |  |
| I, 4, [3],12: i: - | 2 |  |  |  |  |  |  |  | ■ | ■ | ■ | ■ |  | ■ | ■ |
| Altona | 1 |  |  |  |  |  |  |  | ■ | ■ | ■ | ■ |  | ■ | ■ |
| I, 4, [3],12: i: - | 3 |  |  |  |  |  |  |  | ■ | ■ | ■ | ■ |  |  |  |
| I, 4, [3],12: i: - | 3 | ■ |  |  |  |  |  |  | ■ | ■ | ■ | ■ |  |  |  |
| I, 4, [3],12: i: - | 2 | ■ |  |  |  |  |  |  | ■ | ■ | ■ | ■ |  | ■ |  |
| Altona | 1 |  |  |  |  |  |  |  | ■ |  | ■ |  |  |  |  |  |
| Altona | 1 |  |  |  |  |  |  |  | ■ |  | ■ |  | ■ |  |  |  |
| I, 4, [3],12: i: - | 1 | ■ |  |  |  |  |  | ■ | ■ | ■ | ■ | ■ | ■ | ■ | ■ |
| Typhimurium | 1 |  | ■ | ■ |  | ■ |  |  | ■ | ■ | ■ |  |  |  |  |
| I, 4, [3],12: i: - | 1 |  |  |  |  |  |  | ■ |  |  | ■ | ■ |  |  |  |
| Typhimurium | 1 | ■ |  |  |  |  |  |  |  |  | ■ | ■ |  | ■ |  |
| Typhimurium | 1 | ■ |  |  |  |  |  | ■ |  |  |  |  |  |  |  |
| Typhimurium | 1 | ■ |  |  | ■ |  |  |  |  |  |  |  |  |  |  |
| Typhimurium | 1 |  |  |  |  |  |  |  |  |  | ■ |  |  |  |  |

[a]: ampicillin, [b]: cefalexin, [c]: cephalothin, [d]: ceftiofur, [e]: cefoperazone, [f]: gentamicin, [g]: cefquinome, [h]: flumequine, [i]: enrofloxacine, [j]: tetracycline, [k]: florfenicol, [l]: trimetho-prim, [m]: marbofloxacine, [n]: neomicyn.

instructions. Minimum inhibitory concentration (MIC) was interpreted according to Clinical and Laboratory Standards Institute (CLSI) 2020 guidelines [20].

## Whole Genome Sequencing

An overnight culture from a single colony of each *Salmonella* isolate was prepared in 1 ml of lysogeny broth (LB). The DNA from each *Salmonella* isolate was extracted using the PureLink Genomic DNA Mini Kit (Invitrogen, USA) according to the manufacturer's instructions. The purity was assessed by measuring the absorbance ratio at OD 260/280 NanoDrop (Thermo Fisher Scientific), and the yield was established with a Qubit 4.0 Fluorometer (Thermo Fisher Scientific, USA). The library preparation was made using NEBNext Ultra II FS DNA Library Prep with Sample Purification Beads (New England Biolabs, UK) according to the manufacturer's instructions. Immediately, libraries were pooled, quantified, and normalized down to 4nM using the Tape Station System (Agilent, USA). Subsequently, the denaturation library was loaded onto an Illumina MiSeq (San Diego, USA) reagent cartridge using a reagent kit v2 and 300 cycles (150 bp paired-end sequencing) with a Standard Flow Cell. All isolates sequenced had a 50X coverage, <=150 contigs, an average GC content of 51%, an N50 score >50000, and a total assembly length between 4.5–5.2 Mega base pairs (Mbp).

## Genome analysis

The sequencing products were processed using an automatic pipeline from GHRU - The Centre for Genomic Pathogen Surveillance (https://gitlab.com/cgps/ghru/pipelines). The first

pipeline is for assembly and includes the read quality assessed with FastQC v0.11 [21], identification of species with BactInspector v0.1.3 [22] and contamination using Confidr v0.7.2 [23]. The sequences were trimmed and assembled using Trimommatic v0.39 [24], Mash v2.2 [25], Seqtk v1.3 [26], FLASH v1.2.1 [27], Ligther v1.1 [28], and SPAdes v3.14.0. [29], in that order. The assembly's quality was determined with the software Quast v5.0.2 [30]. All running parameters were set as default, expect trimommatic min_length, set as 30% of raw reads size, all databases in docker containers, as described in the official pipeline page (https://gitlab.com/cgps/ghru/pipelines/dsl2/pipelines/assembly). Good quality assemblies were considered when fastqc read quality was not "FAIL" in any of the evaluated parameters, bactinspector showed species name as expected for the isolate, no contamination was reported form Confindr, and assembly quality had less than 150 contigs per assembly with an N50 above 50000 bp.

Prediction of AMR was performed in the second pipeline that uses ARIBA [31] against ResFinder 4.0. [32], PlasmidFinder v2.1 [33], and VFDB [34] respectively (https://gitlab.com/cgps/ghru/pipelines/dsl2/pipelines/amr_prediction). For MSLT prediction MLST ARIBA software was used along with PubMLST databases [35]. All the software used in this automatic pipeline were run with default parameters, their databases were in Docker containers and updated to January 2022. *In silico* serovar prediction, was performed by SISTR v1.1.1 [36] and SeqSero2 v1.2.1. [37] packages with default parameters.

Additional core genome phylogenetic analysis was conducted using EnteroBase [38] on August 11th, 2023. Short-read sequences were assembled using the EnteroBase ToolKit (EtoKi) [38]. The genomic relatedness of the study strains to publicly available isolates on Enterobase was visualized with a GrapeTree plot with filters on the core genome multilocus sequence types (cgMLST) [39]. Swine isolates of serovars Typhimurium and 1, 4,[3],12:i:- from this study were compared to isolates of these serovars collected in Colombia (any source) and to isolates collected from swine (any location).

### Virulence factor identification

For this analysis, the virulence factor database was used (VFDB; https://www.ncbi.nlm.nih.gov/pmc/articles/PMC8728188/). For the final analysis serovar Typhimurium isolate LT2 was used as a reference, as serovar Altona (n = 3) is not present on this database, and serovar I 1,4,[3],12:i:- (n = 20) should match closely with Typhimurium (n = 11). Two isolates (1802 and 1803) were removed from the VF analysis as they had poor quality sequencing metrics.

To compare the genomes to this database, SPADES was used to assemble the shot reads. Then BLAST compared these sequences to the VFDB database, where the query coverage was set to 75% and the percent nucleotide identity was set to 80%.

## Results

### Origin and Identification of strains and serotypes

A total of 653 swine samples were collected between 2019 and 2021 from farms across various municipalities in Antioquia, with 22.8% (149/653) testing positive for *Salmonella*. From the positive samples (n = 149), we selected 28 representative isolates based on antimicrobial resistance profiles observed during phenotypic evaluation while the samples were processed in our diagnostic facility. The goal of this study was to assess the utility of genomics analysis for investigating Salmonella from swine. Of the 28 samples, 8 were gut tissue, 15 were fecal samples, and 5 were rectal swabs. The distribution of these isolates across towns was as follows: Don Matías (n = 7), Amalfi (n = 3), Angelópolis (n = 4), Caldas (n = 2), Entrerríos (n = 1), Medellín (n = 1), San Pedro de los Milagros (n = 1), Santa Rosa de Osos (n = 6), Santo Domingo (n = 1), Támesis (n = 1), and

Valparaíso (n = 1) (see Fig 1). Whole genome sequencing analysis identified three serotypes: *Typhimurium* (n = 10, 36%), *Altona* (n = 3, 11%), and *I 4,[3],12:i:-* (n = 15, 53%).

## Comparison with global serovar Typhimurium isolates

GrapeTree plots were used to visualize whether the serovar Typhimurium isolates from this study are related to 2265 serovar Typhimurium isolates also from swine that are publicly available (S1 Fig). Of the 11 serovar Typhimurium isolates from this study, two lineages were identified. The major lineage included ten isolates tightly clustered with one another and a median pairwise allelic difference (PAD) of 65 from the nearest common isolate. The other lineage included the remaining single serovar Typhimurium isolate (isolate 4351) and had a PAD of 19 from the nearest common isolates from Chile, Ecuador, and the United States. We next looked at serovar Typhimurium isolates from Colombia that were available on Enterobase and compared these to our swine isolates (Fig 2). A total of 88.6% (164/189) were from humans, and only a few non-human isolates were

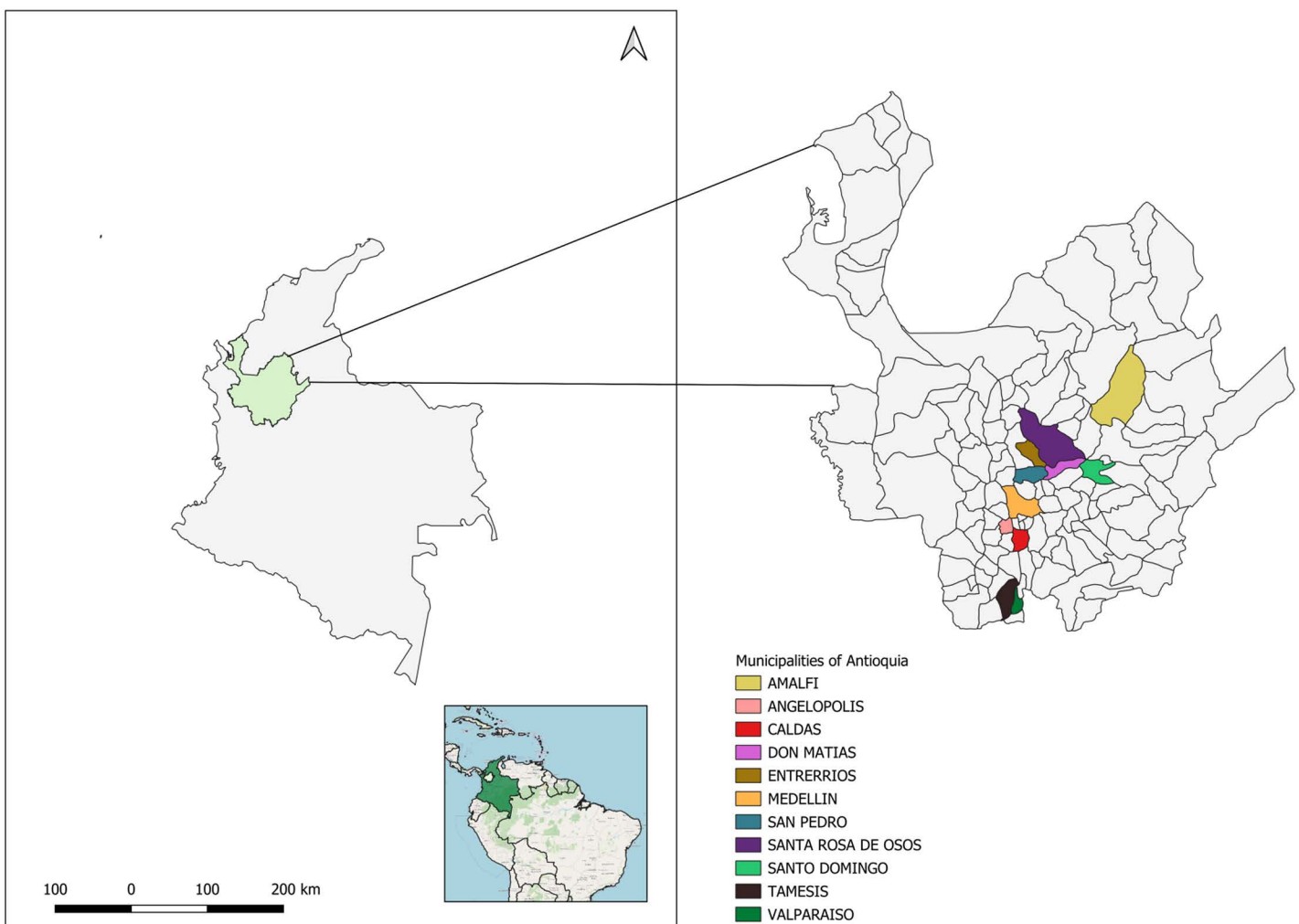

**Fig 1. The geographical map depicts the location of Salmonella isolates in the different municipalities of Antioquia, Colombia.** The base map shapefile used in Qgis 3.34 was sourced from the publicly database Colombia en Mapas (https://www.colombiaenmapas.gov.co/).

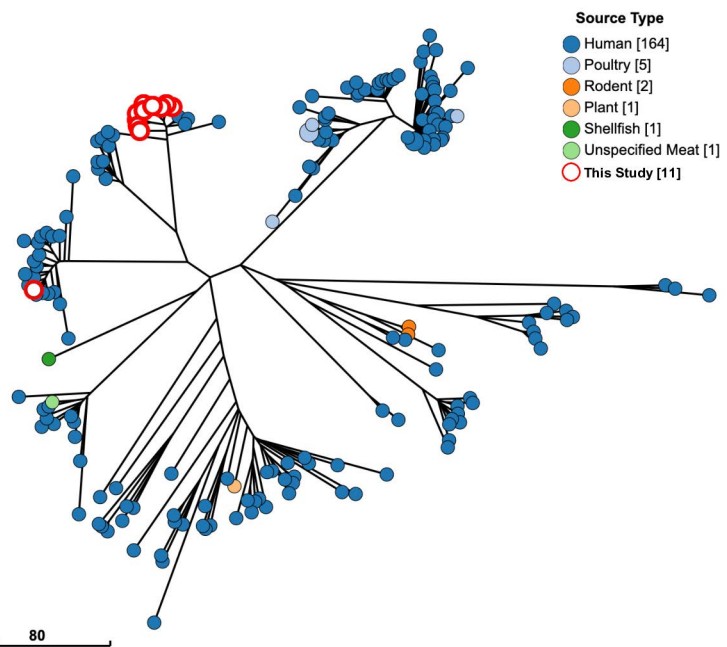

**Fig 2. Serovar Typhimurium isolates from Colombia are most similar to human isolates.** All Salmonella serovar Typhimurium isolates from Colombia in the Enterobase database are shown on a GrapeTree plot using cgMLST profiles to show phylogenetic distance. White circles highlighted in red are isolates from this study. (Scale bar) Number of cgMLST pairwise allelic differences (PADs). "Missing" on the figure legend indicates the source type was not included in the metadata for included samples.

present in this analysis, including poultry (n = 5), rodent (n = 2), plant (n = 2), shellfish (n = 1), and unspecified meat (n = 1). Notably, none were from swine except for the isolates in the current study. All 11 isolates were clustered tightly with human isolates (PAD of < 15), especially isolate 4351, which clustered within a PAD of < 5, indicating a high relatedness to these isolates.

## Comparison of serovar I 4, [3],12: I; - isolates from swine

A total of 2443 swine isolates matching to I,4, [3],12: I; - were plotted on a GrapeTree plot for visualizing phylogenetic analysis (Fig 3). Isolates from this study formed two distinct lineages, where 18 isolates clustered together nearest to seven isolates from Ecuador (PAD of 5). The remaining two isolates that formed the second, more minor lineage, was within a PAD of 10 from each other and five swine isolates from Brazil.

## Comparison of serovar I 4, [3],12: I; - isolates from Colombia

Enterobase contains 32 isolates of serovar I 4, [3],12: I; - isolated from Colombia, and 20 isolates identified in this study. These 52 isolates had phylogenetic relationships visualized by a Grape Tree plot, showing core genome MLST identities (Fig 4). This GrapeTree plot resulted in two lineages of isolates from this study, where 18 clustered within 15 PADs of one another and other human isolates. The remaining two isolates (1638 and 4843) comprised the second, more minor lineage with PADs of less than four from two different human isolates. Notably, all isolates of serovar I 4,[3],12:I;- collected from Colombia outside this project have been from humans.

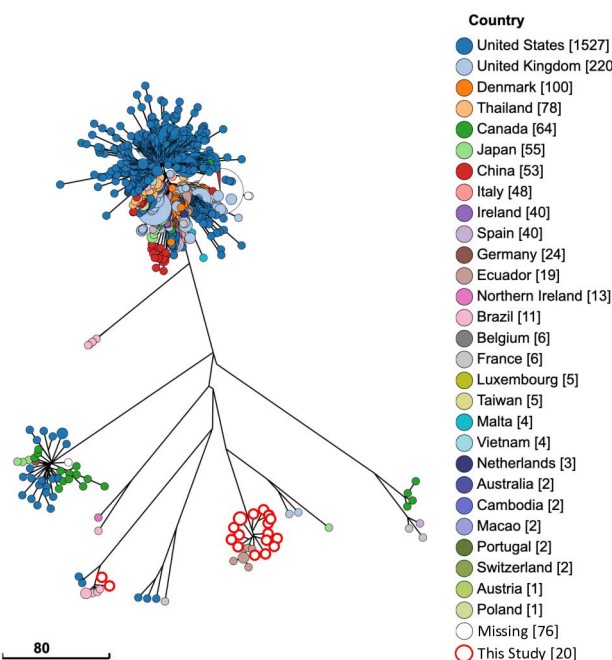

**Fig 3. Serovar I, 4,[5], 12:i:-isolates from swine are closely related to isolates from Ecuador and Brazil.** All Salmonella serovar I, 4,[5], 12:i:- isolates from swine in the Enterobase database are shown on a GrapeTree plot using cgMLST profiles to show phylogenetic distance. White circles highlighted in red are isolates from this study. (Scale bar) Number of cgMLST pairwise allelic differences (PADs). "Missing" on the figure legend indicates the country of sample origin was not included in the metadata for included samples.

## Phenotypic characterization of antimicrobial resistance

All 28 isolates were resistant to at least one antimicrobial (Table 1), and 15 different resistance patterns were found in the 28 isolates. Of these, 26 (93%) had an MDR profile. The most frequently observed MDR profile was AMP-CEX-CEP-CEF-EFT-CEQ-FLU-ENR-TE-FFC-SXT, present in five (18%) serotype Typhimurium isolates. This profile includes six different classes of antibiotics. For serotype I 4, [3],12: i: - the most frequent pattern observed was FLU-TE-FFN (n = 4, 14%), and in the serotype Altona the pattern with the highest number of resistant antibiotics was NEO-FLU-ENR-MAR-TE-FFC (n = 1,3,6%).

## Identification of antimicrobial resistance genes

The complete sequencing of the 28 *Salmonella* isolates revealed a great diversity of AMR genes, as seen in Table 2. In 25 isolates (89%), more than two resistance genes were found. Whereas in only three isolates (11%), only one resistance gene was present. In total, genes encoding resistance were found for ten different groups of antibiotics (beta-lactams, aminoglycosides, chloramphenicol, rifampicin, lincosamidas, fluoroquinolones, tetracyclines, sulfonamides, and trimethoprim) which will be described below.

## Genotypic resistance to beta-lactams

Genes encoding resistance to extended-spectrum betalactamases (ESBL) were found in 17 (61%) isolates. ${}^{bla}$TEM was the most common gene (n = 14, 50%), followed by the ${}^{bla}$OXA-1 genes (n = 6, 21%), ${}^{bla}$OXA-10 (n=3, 10%), ${}^{bla}$OXA (n = 1, 3,5%) and ${}^{bla}$CTX-M (n = 4, 14%). A total of 8 isolates presented combinations of the genes ${}^{bla}$ TEM, ${}^{bla}$OXA-1, and ${}^{bla}$CTX-M (Table 2).

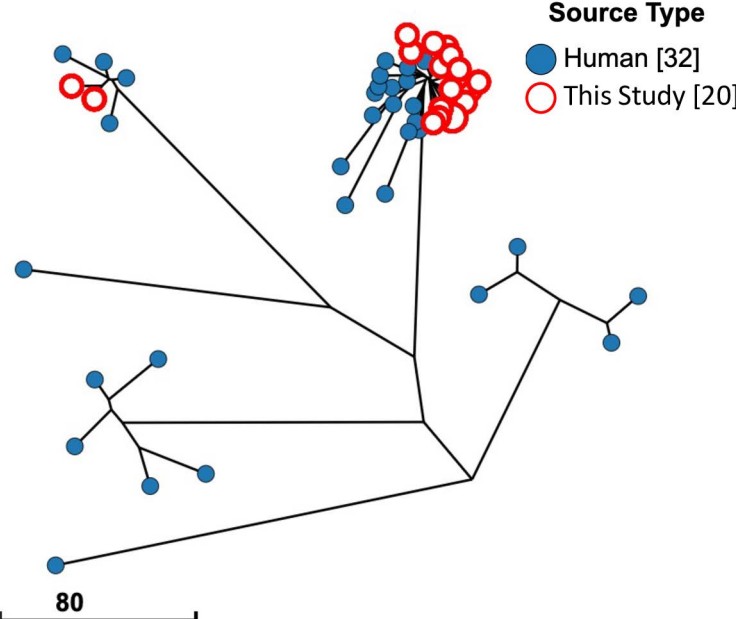

**Fig 4. Serovar I, 4[5], 12:i:- isolates from Colombia show divergence of two clusters.** All Salmonella serovar 1, 4[5], 12:i:- isolates from Colombia in the Enterobase database are shown on a GrapeTree plot using cgMLST profiles to show phylogenetic distance. White circles highlighted in red are isolates from this study. (Scale bar) Number of cgMLST pairwise allelic differences (PADs). "Missing" on the figure legend indicates the source type was not included in the metadata for included samples.

## Genotypic resistance to aminoglycosides

The most remarkable diversity of AMR genes was found for those encoding resistance to aminoglycosides. The gene *aac(3)* was detected in a single isolate (3.5% in the same way as its variations *aac(3)-II* (n = 1, 3.5%) and *aac(3)-Iid* (n = 1, 3.5%) and in a total of 7 isolates (25%) the gene *aac(6')lb-cr* was evidenced. The following genetic cases were observed: *aadA* (n = 2.7%), *aadA1* (n = 17, 61%), *aadA13* (n = 8, 28%), *aadA2* (n = 6, 21%), aadA 12 (n =1, 3.5%). The *aph(3')-lb* gene was found in 11 isolates (39%), followed by the *aph(3')-la* (n = 5, 18%) and *aph(3')-lla* (n = 1, 3.5%) genes. Additionally, the *aph(6)-ld gene* (n = 11, 39%) was also detected (Table 2).

## Genotypic resistance to chloramphenicol, rifampicines, lincosamides, and fluoroquinolones

The *floR* gene (n = 21, 75%) was widely found among the isolates. Other genes that confer resistance to chloramphenicol were also detected: *catA1* (n = 9, 32%), *catB3* (n = 7, 25, and *cmlA* (n = 7, 25%). Regarding resistance to rifampicin, the *arr* gene was found (n = 4, 14%). In the Resistance to incosamidas, the cassette gene *InuF* was evidenced in only one isolate (3.5%). On the other hand, associated with resistance to fluoroquinolones, the *qnRb* gene was found in most isolates (n = 22, 78%) (Table 2).

## Genotypic resistance to tetracyclines, sulfonamides and trimethoprim

The *tetA* gene that confers resistance to tetracyclines was the most found among the isolates (n = 24, 86%). The presence of the *tetB* (n = 1, 3.5%) and *tetM* (n=1, 3.5%) genes was less

**Table 2. AMR genes combinations and phenotypic resistance patterns in *Salmonella* serovars recovered from swine production in Antioquia, Colombia.**

| *Salmonella* strain | Serovar | AMR gene profiles | Phenotypic resistance patterns |
|---|---|---|---|
| 1507 | Typhimurium | *aac(6')lb-cr, aph(3')-lb, aph(6)-ld, aadA, [bla] CTX-M, [bla] OXA-1, [bla] TEM, catA 1, catB 3, dfrA14, floR, sul2, qnRb, tet (A)* | AMP, CEX, CEP, CEF, CFPZ, ENR,TE, FFC, STX, FLU |
| 1804 | Typhimurium | *aac(6')lb-cr, aph(6)-ld, bla OXA-1, [bla] TEM, catA 1, dfrA14, tet (A),sul2* | AMP, TE, SXT |
| 1805 | Typhimurium | *aac(6') lb-cr,aadA1,aph(3')-lb, aph(6)-ld, bla OXA-1, [bla] TEM, catA 1, catB 3, dfrA14, sul2, tet (A)* | AMP, CEF, TE, SXT |
| 1806 | Typhimurium | *aac (6') lb-cr, aadA1, aph(3')-lb, aph(6)-ld,bla OXA-1, [bla] TEM, catA 1, catB 3, dfrA14, sul2, tet (A)* | AMP, TE, SXT |
| 1808 | Altona | *qnRb* | TE, SXT, FLU |
| 2502 | I,4, [3],12: i: - | *aadA1, aadA2, aph(3')-lb, cmlA, dfrA12, floR, sul3, tet (A)* | AMP, ENR, TE, SXT, FLU |
| 3108 | I, 4, [3],12: i: - | *aac(3)-ll, aadA2, aph(3')-lla, aph(3')-la, [bla] TEM, cmlA, dfrA12, floR, Inu(F)* | AMP, GEN, NEO, TE, FFC, SXT, FLU |
| 3469 | I, 4, [3],12: i: - | *floR,qnRb, tet (A)* | TE,FFC, FLU |
| 3977 | Typhimurium | *aac(6')lb-cr, aadA, aadA13, aph(3')-lb, aph(6)-ld, [bla] CTX-M, [bla] OXA-1, [bla] TEM, catA 1, catB 3, dfrA14, floR, sul2, tet (A)* | CEP, CFPZ, CEF, TE, FFC, SXT, FLU |
| 4320 | I, 4, [3],12: i: - | *aph(3')-lb, floR, qnRb, tet (A)* | TE, FFC, FLU |
| 4351 | Typhimurium | *aph(3')-lb, aph(6)-ld, sul2, tet (A)* | TE |
| 4416 | I, 4, [3],12: i: - | *floR,qnRb, tet (A)* | TE, FFC, FLU |
| 4452 | I, 4, [3],12: i: - | *aadA1, aadA13, arr, bla OXA-10, cmlA, dfrA14, floR, qnRb, tet (A)* | AMP, ENR, MAR, TE, FFC, FLU |
| 4594 | Typhimurium | *aac(6')lb-cr, aadA, aadA13, aph(3')-lb, aph(6)-ld, [bla] CTX-M, [bla] OXA-1, [bla] TEM, catA 1, catB 3, dfrA14, floR, sul2, tet (A)* | AMP, CEX, CEP, CFPZ, CEF, CFQ, ENR, TE, FFC, SXT, FLU |
| 4843 | I, 4, [3],12: i: - | *aac(3)-ll, aadA2, aph(3')-lla, aph(3')-la, [bla] TEM, dfrA12, floR, tet (B)* | AMP, ENR, MAR, TE, FFC |
| 5226 | I, 4, [3],12: i: - | *aadA2, dfrA12, floR, qnRb, tet (a)* | TE, FFC, SXT, FLU |
| 5672 | I, 4, [3],12: i: - | *aadA13, aadA1, arr, bla OXA-10, cmlA, dfrA14, floR, qnRb, tet (A)* | AMP, ENR, TE, FFC, FLU |
| 5678 | I, 4, [3],12: i: - | *aadA13, aadA1, [bla] OXA-10, cmlA, dfrA14, floR, qnRb, tet (A)* | AMP, ENR, TE, FFC, FLU |
| 5781 | I, 4, [3],12: i: - | *aadA1, aadA2, bla TEM, cmlA, floR, qnRb, sul3, tet (A)* | AMP, CEF, TE, FFC, FLU |
| 5782 | I, 4, [3],12: i: - | *aadA1, aadA2, [bla] TEM, cmlA, floR, qnRb, sul·, tet (A)* | AMP, CEF, TE, FFC, FLU |
| 5785 | I, 4, [3],12: i: - | *aadA13, aadA1, aadA2, bla TEM, cmlA, floR, qnRb, sul3, tet (A)* | AMP, CEF, TE, FFC, FLU |
| 6113 | Altona | *qnRb* | FLU, TE, FLU |
| 6114 | I, 4, [3],12: i: - | *qnRb* | FLU, ENR, TE, FFC, NEO, FLU |
| 6245 | I, 4, [3],12: i: - | *aadA1, aadA2, aph(3')-la, floR, qnRb, tet (A)* | NEO, ENR, MAR, TE, FFC, FLU |
| 6246 | Altona | *aadA2, aph(3')-la, floR, qnRb, tet (A)* | NEO, ENR, MAR, TE, FFC, FLU |
| 6322 | Typhimurium | *aadA13, aph(3')-lb, aph(6)-ld, bla TEM, catA 1, dfrA14, floR, qnRb, sul2, tet (A)* | AMP, CEX, CEP, CEF, CFPZ, ENR,TE, FFC, STX, FLU |
| 6323 | Typhimurium | *aadA13, aadA1, aph(3')-lb, aph(6)-ld, [bla] TEM, catA1, dfrA14, floR, qnRb, sul2, tet (A)* | AMP, CEX, CEP, CEF, CFPZ, ENR,TE, FFC, STX, FLU |
| 6667 | Typhimurium | *aac(3)-ll, aac(6')lb-cr, aadA13, aadA1, aph(3')-lb, aph(6)-ld, [bla] CTX-M, [bla] OXA, [bla] TEM, catA1, catB 3, dfrA14, qnRb, tet (A)* | AMP, CEX, CEP, CEF, CFPZ, ENR,TE, FFC, STX, FLU |

AMP: ampicillin, CEX: cefalexin, CEP: cephalothin, CFPZ: cefoperazone, CEF: ceftiofur, CFQ: cefquinome, GEN: gentamicin, FLU: flumequine, ENR: enrofloxacine, TE: tetracycline, FFC: florfenicol, SXT: trimethoprim-sulfamethoxazole, MAR: marbofloxacine, NEO: neomicyn.

prevalent. For sulfonamide resistance, the *sul2* (n = 12, 43%), *sul3* (n = 5, 18%) and *sul1* (n = 1, 3.5%) genes were present. Additionally, two genes that confer resistance to trimethoprim were detected: *dfrA14* (n = 12, 43%) and *dfrA12* (n = 4, 14%) (Table 2).

## Virulence genes

In all serovar Altona isolates (n = 3), genes responsible for adherence and effector delivery systems, including the *stcABCD*, *safABCD*, and *pefABCD* operons were absent. Additional virulence factors that were absent only in Altona included: *flgB* (motility), STM3026 (adherence, std cluster), STM0290 (sciW; effector delivery), STM0283 (effector delivery/secretion), *gogB* (effector delivery), *sseI/srfH* (effector delivery), *ratB* (adherence), *mig-5* (antimicrobial), *sspH2* (effector delivery), *sodCI* (stress survival), and *lpfD* (adherence). Unsurprisingly, Typhimurium and I 4,[3],12:i:- isolates have very similar virulence factors. The main difference is the expected absence of *fljA* and *fljB* motility genes in I 4,[3],12:i:-. The *sspH1* gene was absent from all 28 genomes.

## Discussion

In Colombia, several studies have documented that the prevalence of Salmonella spp. In pig farms exceeds 25% [40–43]. Antioquia, as the leading pig producer in the country, has experienced relatively low disease prevalence, primarily due to advancements in technology and enhanced biosecurity measures on swine farms [42]. However, it is important to note that there are currently no regulations or action plans for antimicrobial use in pig production, nor strategies for reducing antimicrobial use in Colombia.

Among the isolates evaluated, three serotypes were found: Typhimurium, I 4, [3],12:i:-, and Altona. Serotype Typhimurium is frequently found in pigs, slaughterhouses, and pork products [44] and is one of the most prevalent serovar worldwide [45]. It is responsible for 17.4% of human salmonellosis in the European Union, North America, and Oceania, but this differs in South America, where the Meleagridis serovar is the most reported in the region [4,46]. *Salmonella* I, 4, [3],12: i:-, a monophasic variant of serotype Typhimurium, is one of the five most commonly reported serovars in human diseases worldwide [47]. Additionally, this serovar is widely associated with swine production, especially in Europe and the United States, suggesting a potential link between human infections and consumption of contaminated pork products [48]. *Salmonella* Altona was the least found serotype. An outbreak in humans associated with this serotype was reported in the United States, affecting 68 people across 20 states in 2011 [49], where the risk from contact with live poultry was highlighted, especially for young children [50]. This is the first study to report the presence of serotype Altona in Colombia.

In this study, patterns like the ACSuGSTTm (ampicillin, chloramphenicol, sulfonamides, gentamicin, streptomycin, tetracycline, and trimethoprim) predominated among serotype I, 4, [3],12: i:-, isolates, representing 67% (10/15) of the cases. Globally, serotype I, 4, [3], 12: i:-, has displayed three prominent lineages, all associated with swine but exhibiting distinct antimicrobial resistance profiles. The "Spanish clone" presents the ACSuGSTTm phenotypic pattern [51], whereas the "USA clone" lacks an extensive MDR pattern but shows resistance to quinolones and extended-spectrum cephalosporins [52]. The "European clone" has been documented in multiple EU countries, characterized by a predominant ASSuT pattern (ampicillin, streptomycin, sulfonamides, and tetracycline) [53].

The highest rate of phenotypic resistance is recorded for tetracyclines (100%), which is one of the most widely used antimicrobials in pig feed in Colombia. In commercial pig operations, tetracyclines are used for therapeutic purposes to treat septicemia and intestinal and respiratory infections [54,55]. Resistance to tetracyclines is frequent in both commensal and pathogenic

bacteria, primarily because of the acquisition of [bla]*tet* genes [56–58]. In this study, a high presence of the [bla]*tetA* gene was found (86%), concordant with multiple investigations in swine conducted in China, where the gene was present in 64% and 95% of the isolates evaluated [59,60].

The production of beta-lactamases is considered the primary mechanism of resistance to beta-lactam antibiotics [61], especially in gram-negative pathogens [62]. The literature describes that *Salmonella* isolates harboring ESBL enzymes have emerged globally recently, with the [bla]CTX-M [63] group being critical. Despite this, the [bla]TEM gene was the most found in this study (50%), and the [bla]CTX-M gene was only evidenced in 14% of the isolates.

Prior investigations have documented that chloramphenicol resistance primarily arises through inactivation by chloramphenicol acetyltransferase (CAT), which is encoded by the *catA1*, catA2, and *catB* genes [64]. Although these genes were identified in our study, they did not exhibit high prevalence. Instead, our study reveals the prominence of the *floR* gene (75%) as the principal factor in conferring chloramphenicol resistance. This gene has been previously documented in serotype Typhimurium and its monophasic variant (I 4,[3],12:i:) [65], with the floR gene assuming a pivotal role within the cohort of multidrug resistance genes concomitant with SGI1 [66].

Despite their toxicity, and the residues in tissues, antibiotics such as aminoglycosides, have been used to treat undifferentiated diarrhea in weaned piglets and swine dysentery caused by *Brachyspira hyodysenteriae* [67]. In this study, the *aph* genes were identified in most isolates, underscoring a notable genetic diversity with a total of 6 distinct gene. Most *aph(3')* enzymes are widespread among pathogenic microorganisms and, together with the enzyme *aph(6)* (also found in this study), are highly related to resistance to streptomycin [68]. Two recent studies conducted in Brazil revealed the presence of the *aph(3')-lb* gene in samples of humans and contaminated food [69,70]. The *aph(3')-lb* gene was the most commonly associated with aminoglycoside resistance in this study.

The *qnRb* gene, conferring resistance to the quinolones, was widely distributed among our isolates, and was found in all the three serotypes. In contrast, studies conducted in Thailand and China showed a low distribution of the gene among the isolates analyzed [71,72]. This gene imparts reduced resistance levels to quinolones [73], and resistance is directly related to the expression levels of gene [74]. On the other hand, resistance to trimethoprim is mediated by genes encoding dihydrofolate reductase variants (*dhfr* and *dfr*) that have decreased affinity for the antimicrobial agent [75]. The most found gene was *dfrA14*, previously reported in *Salmonella* isolates from raw chicken in Brazil, Chile, and Thailand [76].

In veterinary medicine, the development of antimicrobial resistance is closely related to the excessive use of antimicrobial agents, which facilitates microorganisms with newly acquired mutations or resistance genes to survive and proliferate [77]. Evidence shows that food from animal sources at all stages of food processing contain many resistant bacteria [78]. The concern focuses on the horizontal transmission of genes, being the main responsible for the transmission of AMR, which results in a rapid dissemination of AMR between bacteria and species [79]. Globally, a crisis unfolds as we repeatedly observe a gene in a specific organism, subsequently leading to global reports of the gene appearing in other organisms and locations [80].

Reduced and prudent use of antimicrobials in animal production could limit the selection of resistant bacteria and may lead to a reversal of susceptibility [81]. Countries such as Denmark have made significant progress in controlling AMR by implementing management programs in the veterinary sector as early as 1998. As a result, in Denmark, the amount of antimicrobials consumed per kilogram of produced pork fell by 50% between 1994 and 2013 [82]. In Colombia, there is a need for improved and controlled use of antibiotics and surveillance strategies to help reduce the high levels of AMR observed in production animals [83]. This study illustrates that the Salmonella isolates assessed in pigs harbor a diverse array

of antimicrobial resistance genes. This is also a problem for veterinarians as the emergence of AMR and MDR can increase the spread of resistant pathogens within animal production systems causing treatment failures that can put animal health at risk [84]. The contribution of individual countries in reducing AMR will be essential to mitigate the global problem since antibiotic resistance spreads on a macroscopic scale. Antimicrobial resistance genes (ARGs) in plasmids can facilitate the spread from a resistant bacterium to a susceptible one, allowing the development of new resistant organisms. On the other hand, individual resistant organisms and colonies spread around the world as colonized or infected patients travel. The mode of spread is on the rise with the increasing globalization [85].

An earlier molecular characterization conducted in serovars Typhimurium and 1,4,[3],12:i:- found the *sspH1* gene in 77% of isolates as a virulence determinant related to the prophage [86], a finding not observed in this study. Additionally, previous studies found an association between the presence of the *sspH1* gene and phenotypic MDR [87]. However, this association was not identified in our study, as the presence of the sspH1 gene was not detected in any of the isolates.

The phylogenetic comparison of isolates from this study to other isolates from swine or Colombia was limited due to the low number of swine isolates from Colombia and elsewhere in South America compared to other countries, such as the United States, where whole genome sequencing is performed more routinely. The few publicly available isolates present were found to be genetically like to the isolates of this study, suggesting that strains of both serovars Typhimurium and I 4, [3],12: i: - could be commonly identified in this region. There was no concordance between AMR profiles in the more minor lineages containing only one or two isolates when compared to the remaining study strains of either serovar Typhimurium or I 4, [3],12: i: - but further analysis is required to determine the genomic differences between clades observed on the GrapeTree plots for both serovars. Due to a limited number of studies investigating the burden of *Salmonella* in Colombia [88], human isolates make up most data on Enterobase, which limits the information that can be obtained through the newly acquired WGS data presented here. Thus, doing regular surveillance of *Salmonella* from swine production in Colombia, increasing the number of comparable isolates and identifying strains related to human outbreaks, could increase the efficiency in detecting related isolates and in supporting targeted and improved *Salmonella* control for swine producers.

The limitation of resources in this study precluded the phenotypic and genome sequencing analysis of all Salmonella isolates. The authors believe that employing such technology would be of significant value for molecular epidemiology studies. These would enhance the understanding of the dynamics of antimicrobial resistance in Salmonella isolates in Antioquia, Colombia. Furthermore, complete genome sequencing could provide deeper insights into the genetic mechanisms driving resistance, facilitate the identification of novel resistance genes, and support the development of more targeted and effective control strategies. Addressing this limitation in future research could greatly contribute to a more nuanced understanding of resistance patterns and inform public health and policy interventions.

## Conclusion

This study, conducted through whole-genome sequencing, not only identified the presence of multidrug-resistant *Salmonella* in pigs intended for human consumption in Antioquia, Colombia but also allowed for a comprehensive analysis of genetic diversity and precise identification of resistance genes. This advanced approach provided a more detailed and complete understanding of virulence and antimicrobial resistance factors, emphasizing the significance of whole-genome sequencing in comprehensively addressing the issue of resistance in pig

production and its impact. Furthermore, the authors underscore the imperative for the formulation of public policies aimed at regulating antibiotic use within pig production systems in Colombia. They also advocate for the establishment of educational programs designed to enhance producer awareness regarding the rational use of antibiotics.

## Supporting information

**S1 Fig. Serovar Typhimurium isolates from swine form two distinct clusters.** All *Salmonella* serovar Typhimurium isolates from swine in the Enterobase database are shown on a GrapeTree plot using cgMLST profiles to show phylogenetic distance. White circles highlighted in red are isolates from this study. (Scale bar) Number of cgMLST pairwise allelic differences (PADs)."Missing" on the figure legend indicates the country of sample origin was not included in the metadata for included samples.
(TIF)

## Author contributions

**Conceptualization:** Pilar Donado-Godoy, Luis M. Gómez-Osorio, Jenny J. Chaparro-Gutierrez.

**Data curation:** María Isabel García-Alvarez, Jared Smith.

**Formal analysis:** María Isabel García-Alvarez, Jared Smith, Nikki Shariat, Sara López-Osorio.

**Funding acquisition:** Pilar Donado-Godoy, Jenny J. Chaparro-Gutierrez.

**Investigation:** Juana L. Vidal, Nikki Shariat, Sara López-Osorio.

**Methodology:** Juana L. Vidal, María Fernanda Valencia.

**Project administration:** Pilar Donado-Godoy.

**Validation:** María Fernanda Valencia.

**Writing – original draft:** María Isabel García-Alvarez.

**Writing – review & editing:** Juana L. Vidal, Jared Smith, Nikki Shariat, Luis M. Gómez-Osorio, Sara López-Osorio, Jenny J. Chaparro-Gutierrez.

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
