## [Decision Letter · Decision Letter 0]

20 Aug 2024

Dear Professor Chaparro-Gutierrez,

Thank you very much for submitting your manuscript "Genomic characterization of antimicrobial-resistant and virulence factors in Salmonella isolates obtained from pig farms in Antioquia, Colombia" for consideration at PLOS Neglected Tropical Diseases. As with all papers reviewed by the journal, your manuscript was reviewed by members of the editorial board and by several independent reviewers. In light of the reviews (below this email), we would like to invite the resubmission of a significantly-revised version that takes into account the reviewers' comments.

We cannot make any decision about publication until we have seen the revised manuscript and your response to the reviewers' comments. Your revised manuscript is also likely to be sent to reviewers for further evaluation.

Sincerely,

Ahmed Hassan Fahal, FRCS, FRCSI, FRCSG, MS, MD, FRCP(London)

Academic Editor

Stuart Blacksell

Section Editor

Reviewer's Responses to Questions

**Key Review Criteria Required for Acceptance?**

**Methods**

-Are the objectives of the study clearly articulated with a clear testable hypothesis stated?

-Is the study design appropriate to address the stated objectives?

-Is the population clearly described and appropriate for the hypothesis being tested?

-Is the sample size sufficient to ensure adequate power to address the hypothesis being tested?

-Were correct statistical analysis used to support conclusions?

-Are there concerns about ethical or regulatory requirements being met?

Reviewer #1: 1. Objectives and Hypothesis:

The aim of the study is clearly stated and no hypothesis is stated.

2. Study Design:

This design used was suitable for the purposes of achieving the intended objectives. However, the lack of a clear inclusion and exclusion criteria raises concerns about inference and generalisation of their findings.

3. Population Description/ sample size

The sample size determination is not clearly outlined, neither is the inclusion and exclusion criteria.

statistical analysis

Unfortunately, the submission is lacking in this aspect, the authors fail to describe much of the methodology used during their comparison between genotypic and phenotypic resistance.

4.Ethical or regulatory requirements

There are no concerns about ethical or regulatory requirements not being met. However, it might be helpful for them to include any supporting documents during their submission.

Reviewer #2: -Are the objectives of the study clearly articulated with a clear testable hypothesis stated?

Yes the objectives were well stated following the background about the study

-Is the study design appropriate to address the stated objectives?

Yes

-Is the population clearly described and appropriate for the hypothesis being tested?

Yes

- Is the sample size sufficient to ensure adequate power to address the hypothesis being tested?

There was no clear description of the type of sampling (purposive, random) and sample size estimation/ calculation that was carried out to detect resistant genotypes, and how this was used to obtain these study samples. The authors reported 653 samples collected from 11 towns in 3 years for which 28 were sequenced. The numbers reported here are low given the sampling frame and duration and hence a need to show clear validation from the sample size calculation.

-Were correct statistical analysis used to support conclusions?

Yes

-Are there concerns about ethical or regulatory requirements being met?

In addition of the institutional REC, wasn't their approval from the National research governing council? The details about the consent (sample collection, storage and data publishing) for the study is missing yet worth including in the manuscript.

**Results**

-Does the analysis presented match the analysis plan?

-Are the results clearly and completely presented?

-Are the figures (Tables, Images) of sufficient quality for clarity?

Reviewer #1: With the specific example of the comparison of genotypic and phenotypic resistance resistance, no methodology was given for that particular result. The figures and tables are of sufficient quality.

Reviewer #2: -Does the analysis presented match the analysis plan?

Yes the analysis covers the analysis plan of the study objectives

-Are the results clearly and completely presented?

Yes

-Are the figures (Tables, Images) of sufficient quality for clarity?

Yes

**Conclusions**

-Are the conclusions supported by the data presented?

-Are the limitations of analysis clearly described?

-Do the authors discuss how these data can be helpful to advance our understanding of the topic under study?

-Is public health relevance addressed?

Reviewer #1: The conclusions could be narrowed down to the specific findings of the study.

The limitations of the study are not described.

The authors attempt to discuss how their findings can be helpful in advancing our understanding of the topic under study and the public health relevance is addressed.

Reviewer #2: -Are the conclusions supported by the data presented?

Yes the discussion of the results and conclusion are well supported by the data

-Are the limitations of analysis clearly described?

The limitations on the sample size sequenced with respect to the site coverage was not described and yet this is worth mentioning

-Do the authors discuss how these data can be helpful to advance our understanding of the topic under study?

Yes

-Is public health relevance addressed?

Yes

**Editorial and Data Presentation Modifications?**

Reviewer #1: (No Response)

Reviewer #2: Methodology

Line 116. The statement, “A total of 28 isolates of Salmonella spp. were included”. This is somewhat misleading since the authors proceed with the sentence on “Fecal samples, rectal swabs and tissue samples…” Does this imply that 28 fecal, rectal swab and tissue samples were collected or 28 known isolates of Salmonella spp were added to the samples collected? There is a need to rephrase this paragraph for clarity on the sample collection. In addition, please be more specific about which “tissue” sample was collected, blood, skin? Also good to mention the time/ point of fecal sample collection, was it immediately after defecation? Was it obtained from the rectum?

Line 124. Again, which tissue was collected?

Line 132. It is worth mentioning the biochemical and serological confirmatory tests for Salmonella that were carried out on the isolates.

Results

Line 208. Describe the type of sampling (purposive, random) and sample size estimation calculation that was required to detect resistant genotypes, and how this was used to obtain these study samples. The 653 samples collected from 11 towns in 3 years seems few!

Line 210. The 28 representative samples, it is worth mentioning the sample source as indicated under the methodology sample collection section, for instance how many of these 28 isolates were collected from fecal, rectal swab or tissue.

Line 219. The context under this subsection involved comparison of the Typimurium isolates from this study with global isolates from Enterobase database. I suggest the subsection heading is changed to “comparison with global serovar Typhimurium isolates”

Line 270. Change sub title to “identification of antimicrobial resistance genes”

Line 322. When comparing phenotypic and genotypic resistant isolates, the authors mention correlation analysis however there is no correlation coefficients nor r-squared values given to support the claims. Support the term “discrepancy” with the correlation coefficient values for the comparison.

**Summary and General Comments**

Reviewer #1: * The study should have a clear justification for the sample size determined (i.e is the study adequately powered? was the sampling purposive?).

* The sample inclusion and exclusion criteria should be clearly articulated along with any statistical implications.

* The methodology for any statistical comparisons should be clearly stated (genotypic resistance vs phenotypic resistance).

Reviewer #2: (No Response)

PLOS authors have the option to publish the peer review history of their article (what does this mean? ). If published, this will include your full peer review and any attached files.

**Do you want your identity to be public for this peer review?** For information about this choice, including consent withdrawal, please see our Privacy Policy .

Reviewer #1: Yes: Kimuda Magambo Phillip

Reviewer #2: No
---

## [Editor Report · Decision Letter 1]

21 Oct 2024

Dear Professor Chaparro-Gutierrez,

Thank you very much for submitting your manuscript "Genomic characterization of antimicrobial-resistance and virulence factors in Salmonella isolates obtained from pig farms in Antioquia, Colombia" for consideration at PLOS Neglected Tropical Diseases. As with all papers reviewed by the journal, your manuscript was reviewed by members of the editorial board and by several independent reviewers. In light of the reviews (below this email), we would like to invite the resubmission of a significantly-revised version that takes into account the reviewers' comments.

We cannot make any decision about publication until we have seen the revised manuscript and your response to the reviewers' comments. Your revised manuscript is also likely to be sent to reviewers for further evaluation.

Sincerely,

Ahmed Hassan Fahal, FRCS, FRCSI, FRCSG, MS, MD, FRCP(London)

Academic Editor

Stuart Blacksell

Section Editor
---

## [Decision Letter · Decision Letter 2]

8 Jan 2025

Dear Professor Chaparro-Gutierrez,

We are pleased to inform you that your manuscript 'Genomic characterization of antimicrobial-resistance and virulence factors in Salmonella isolates obtained from pig farms in Antioquia, Colombia' has been provisionally accepted for publication in PLOS Neglected Tropical Diseases.

Best regards,

Ahmed Hassan Fahal, FRCS, FRCSI, FRCSG, MS, MD, FRCP(London)

Academic Editor

Stuart Blacksell

Section Editor

Shaden Kamhawi

co-Editor-in-Chief

Paul Brindley

co-Editor-in-Chief

Reviewer's Responses to Questions

**Key Review Criteria Required for Acceptance?**

**Methods**

-Are the objectives of the study clearly articulated with a clear testable hypothesis stated?

-Is the study design appropriate to address the stated objectives?

-Is the population clearly described and appropriate for the hypothesis being tested?

-Is the sample size sufficient to ensure adequate power to address the hypothesis being tested?

-Were correct statistical analysis used to support conclusions?

-Are there concerns about ethical or regulatory requirements being met?

Reviewer #1: The authors included recommendations such as an ethical approval statement, details on WGS and making their raw sequence data publicly accessible.

**Results**

-Does the analysis presented match the analysis plan?

-Are the results clearly and completely presented?

-Are the figures (Tables, Images) of sufficient quality for clarity?

Reviewer #1: The recommendation below was not adequately addressed. Granted the authors included Table 2 which helps a lot, however the statistical comparisons would add value.

The authors could consider carrying out a correlation analysis between the occurrence of antimicrobial resistance genes (genotypic resistance) and resistant phenotypes from the antimicrobial susceptibility testing (e.g., chi-square tests for categorical data).

**Conclusions**

-Are the conclusions supported by the data presented?

-Are the limitations of analysis clearly described?

-Do the authors discuss how these data can be helpful to advance our understanding of the topic under study?

-Is public health relevance addressed?

Reviewer #1: The conclusion is adequate.

**Editorial and Data Presentation Modifications?**

Reviewer #1: The recommendation below was not adequately addressed. Granted the authors included Table 2 which helps a lot, however the statistical comparisons would add value.

The authors could consider carrying out a correlation analysis between the occurrence of antimicrobial resistance genes (genotypic resistance) and resistant phenotypes from the antimicrobial susceptibility testing (e.g., chi-square tests for categorical data).

**Summary and General Comments**

Reviewer #1: (No Response)

PLOS authors have the option to publish the peer review history of their article (what does this mean? ). If published, this will include your full peer review and any attached files.

**Do you want your identity to be public for this peer review?** For information about this choice, including consent withdrawal, please see our Privacy Policy .

Reviewer #1: No

---

## [Editor Report · Acceptance letter]

Dear Professor Chaparro-Gutierrez,

We are delighted to inform you that your manuscript, "Genomic characterization of antimicrobial-resistance and virulence factors in Salmonella isolates obtained from pig farms in Antioquia, Colombia," has been formally accepted for publication in PLOS Neglected Tropical Diseases.

Best regards,

Shaden Kamhawi

co-Editor-in-Chief

Paul Brindley

co-Editor-in-Chief
